# Filtering Photon Cloud Data in Forested Areas Based on Elliptical Distance Parameters and Machine Learning Approach

Yi Li [1], Jun Zhu [2,*], Haiqiang Fu [1], Shijuan Gao [1] and Changcheng Wang [1]

1    School of Geoscience and Info-Physics, Central South University, Changsha 410083, China; fysxjsxw@csu.edu.cn (Y.L.); haiqiangfu@csu.edu.cn (H.F.); gaoshijuan@csu.edu.cn (S.G.); wangchangcheng@csu.edu.cn (C.W.)
2    School of Traffic and Transportation Engineering, Changsha University of Science and Technology, Changsha 410114, China
*    Correspondence: jzhu@csust.edu.cn

**Abstract:** The Ice, Cloud, and Land Elevation Satellite-2 (ICESat-2) was successfully launched. Due to its small spot size, multibeam configuration, high sampling rate, and strong immunity to terrain slopes, it has been regarded as a powerful tool for forest resources surveying and managing. However, the ICESat-2 photon cloud data contain considerable background photons, which discretely distribute in the background space of signal photons. Therefore, it is necessary to filter these noise photons. In this study, photons are divided into three categories: signal photons, noise photons far away from signal photons, and noise photons adjacent to signal photons. Based on the existing research, forward and backward elliptical distances were used to express the spatial relationship between two photons, and backward local density (*BLD*) was used to describe the density distribution of the photons. However, the single statistical parameter cannot clearly distinguish three types of photon cloud. Therefore, forward local density (*FLD*) and neighboring forward local density difference (*NFLDD*) also were defined to describe the density distribution of the photons. Finally, by combining the support vector machine (SVM), the above three density parameters were used to classify the photons by signal and noise photons. The proposed method was validated with photon cloud data acquired by the Simulated Advanced Terrain Laser Altimeter System (MATLAS), the Multiple Altimeter Beam Experimental Lidar (MABEL), and the ICESat-2 systems over different forested areas. The results demonstrated that the proposed method can well remove the noise photons and retain the signal photons without depending on any statistical assumptions or thresholds. The comprehensive accuracy of the three test sites was 0.99, 0.98, and 0.99, respectively, which was higher than those of the existing method. In addition, the total errors corresponding to the three test sites were about 0.4%, 0.5%, and 1.0% respectively, which were lower than those of the existing method.

**Keywords:** ICESat-2; photon cloud filtering; forested areas; elliptical distance parameters; noise photons; machine learning

## 1. Introduction

The Ice, Cloud, and Land Elevation Satellite-2 (ICESat-2), the successor to ICESat-1, is equipped with the Advanced Terrain Laser Altimeter System (ATLAS) [1]. It produces dense footprints with a 17 m diameter and has the characteristics of high resolution and strong immunity to terrain slopes [2,3]. As a result, ICESat-2 has shown great potential for vegetation parameter extraction and forest management [4]. However, photon cloud data contain considerable background noise photons caused by solar background [5]. In such a case, the removal of these noise photons is very critical for forestry management. In particular, for vegetation areas, photons can penetrate the vegetation layer due to the gaps, and it is more difficult to filter the noise photons. To retrieve accurate vegetation height and sub-canopy topography, a good filter must be employed.

The main principle of filtering photon cloud data relies on differences in the density distributions of the noise and signal photons. Several filtering methods have been proposed, such as the voxel-based filtering method [6], the probability density function-based method [7], the Bayesian decision theory-based method [8], and the density clustering-based method [9–12]. Although the above methods have been successfully tested with the photon cloud data acquired over different terrain conditions with different surface features, they still struggle to completely remove noise photons, especially in forested areas [13]. For complex forested scenes, localized statistics-based filtering methods, such as the local distance statistics (LDS)-based method and the relative neighboring relationship (RNR)-based method [14,15], the modified density-based spatial clustering method [16], and the local outlier factor-based method [17], are more suitable for removing noise photons [13,18]. However, to achieve satisfactory results, localized statistics-based methods must carefully address three main problems: (1) how to design a photon neighborhood search method that can describe the distribution of photons under different terrain conditions, (2) how to define a reasonable filter based on localized statistics that can distinguish signal and noise photons, and (3) how to remove as many noise photons adjacent to signal photons as possible.

A reliable photon neighborhood search method should have the ability to find the neighboring photons around a photon with a similar density distribution, which determines whether the subsequent localized statistics contain enough information to distinguish the signal and noise photons. To achieve this goal, the circle-based search method has been used to define the neighboring photons of a photon [14,15,19]. Although this method can easily find all neighboring photons, it tends to incorrectly select more noise photons [16]. To resolve this issue, the horizontal ellipse-based search method has been used to select neighboring photons [16,17,20]. Compared with the circle-based search method, the horizontal ellipse-based search method can define the neighboring photons of a photon more precisely. However, its performance is strongly dependent on the topographic relief and it does not work well when the topographic relief is obvious. To overcome this limitation, an ellipse with adaptive orientations has been adopted to find the neighboring photons of a photon as accurately as possible under different terrain conditions [13,21].

Using the above neighborhood search methods, the neighboring photons of a photon can be selected, and then various statistical parameters can be used to describe the photon distribution, including the density [16,21,22], the local outlier factor [20], the local distance [14], and the number of neighbors [13,19,20]. The statistics of the signal and noise photons can then be fitted by Gaussian-like functions, and the number of signal photons can be distinguished from background noise photons by setting a reasonable threshold for the statistics. However, selecting an optimal threshold for the statistics that can completely distinguish the noise and signal photons is difficult because the threshold varies with many factors, such as the surface cover, topographic condition, and photon acquisition condition. Moreover, in some cases, these statistics do not follow a strict Gaussian distribution, which increases the difficulty of selecting a suitable threshold. As a result, certain noise photons may not be removed or all the signal photons may not be retained.

In fact, photons can be divided into three categories: signal photons, noise photons far away from signal photons, and noise photons adjacent to signal photons. Even if an optimal threshold is adopted, the existing methods may not completely remove noise photons adjacent to signal photons [19] because noise photons adjacent to signal photons present similar distribution trends as the signal photons. Moreover, single statistical parameter-based methods (the existing localized statistics-based methods) cannot clearly distinguish signal photons and noise photons adjacent to signal photons. Hence, it is necessary to further investigate the distribution characteristics of noise photons adjacent to signal photons and to define a new statistical parameter to describe their densities.

In this paper, a new photon cloud filtering method that combines the proposed elliptical distance parameters and machine learning is proposed. The oriented ellipse-based search method [16,21] is employed to define the neighboring photons around a photon.

Subsequently, the backward local density (*BLD*) [21] is used to define the three category photons. In addition, the forward local density (*FLD*) and the neighboring forward local density difference (*NFLDD*) are then defined to better describe the heterogeneity between signal photons and noise photons. Therefore, the three statistical parameters can better describe the distribution of signal photons, noise photons adjacent to signal photons, and noise photons far away from signal photons, cooperatively. For single statistical indicator-based filtering methods, they distinguish the signal and noise photons by calculating the threshold. However, for multi-parameters, it is unreasonable to directly distinguish signal and noise by the threshold, which increases the error caused by calculating the threshold. The machine learning approach provides a way for multi-statistical parameters to feature fusion. Therefore, with the above three statistical parameters, a machine learning approach is used to distinguish signal photons and noise photons. The proposed method has two main advantages over the existing filtering methods: (1) three statistical parameters are simultaneously adopted to describe the distribution characteristics of the signal and noise photons to better distinguish the noise and signal photons, especially for those noise photons adjacent to signal photons; and (2) machine learning is used to adaptively identify the signal and noise photons without considering the statistical distribution or setting a threshold.

The novelty of this paper is as follows:

1. We define the *FLD* to better distinguish signal photons and noise photons far away from signal photons. In addition, we define the neighboring forward local density difference (*NFLDD*) to retain some signal photons that can be easily recognized as noise photons by the *BLD* parameter.
2. The above two statistical parameters are defined to express the spatial density difference in photon clouds together with the *BLD*. Therefore, different types of photon clouds can be better expressed by three parameters rather than a single parameter.
3. The machine learning approach is used to combine the *FLD*, *BLD*, and *NFLDD* attribute parameters; it is possible to distinguish the noise and signal photons without depending on any statistical model or threshold.

## 2. Materials and Methods

### 2.1. Photon Cloud Datasets

To test the performance of the proposed method on the photon cloud data acquired by various platforms, the proposed method was first tested on the airborne photon data and then verified on the ICESat-2 data. The two kinds of airborne photon data were acquired by MABEL (Multiple Altimeter Beam Experimental Lidar data) sensor and MATLAS (the simulated ATLAS system) data.

2.1.1. Airborne Photon Cloud Datasets and Test Sites

Before the launch of the ICESat-2 satellite, in order to test the quality and performance of the photon cloud data, NASA organized airborne flight experiments on some typical research areas and obtained a series of airborne simulated photon cloud data, such as MABEL and MATLAS data. MABEL photon cloud data are measured by emitting infrared (1046 nm) and green band (523 nm) pulses with a variable repetition rate (5~25 kHz) at a flying height of 20 km, and have a spot diameter of 2 m [23,24]. MATLAS photon cloud data are simulated by the MABEL photon cloud data, which aim to be more similar to the photon cloud data acquired on the ATLAS system [23].

Figure 1 shows the original photon maps of two study areas. Test site A, which is located on the west coast of the USA, is a mountainous area covered with dense vegetation. The corresponding photon data were collected by the airborne MATLAS system, which can be downloaded from http://icesat.gsfc.nasa.gov/icesat2 (accessed on 15 March 2019). Test site B, which is located in Alaska, USA, is characterized by gently sloping topography. The granule ID of these data is MABEL_l2_20140801t173100_010_1, and we obtained MABEL data from http://icesat.gsfc.nasa.gov/icesat2 (accessed on 15 March 2019). The strong

beam is used to test the proposed method, as demonstrated in the existing work [19]. It can be seen from Figure 1 that in comparison to the photon data from test site A, the photon data acquired over test site B include denser and more nonuniformly distributed noise photons. In addition, due to the dense vegetation in test site A, more photons are concentrated in the canopy layer than in test site B, where more photons can reach the sub-canopy ground surface. Test sites A and B are both typical forest areas officially announced by NASA, which can effectively test the performance of the proposed method.

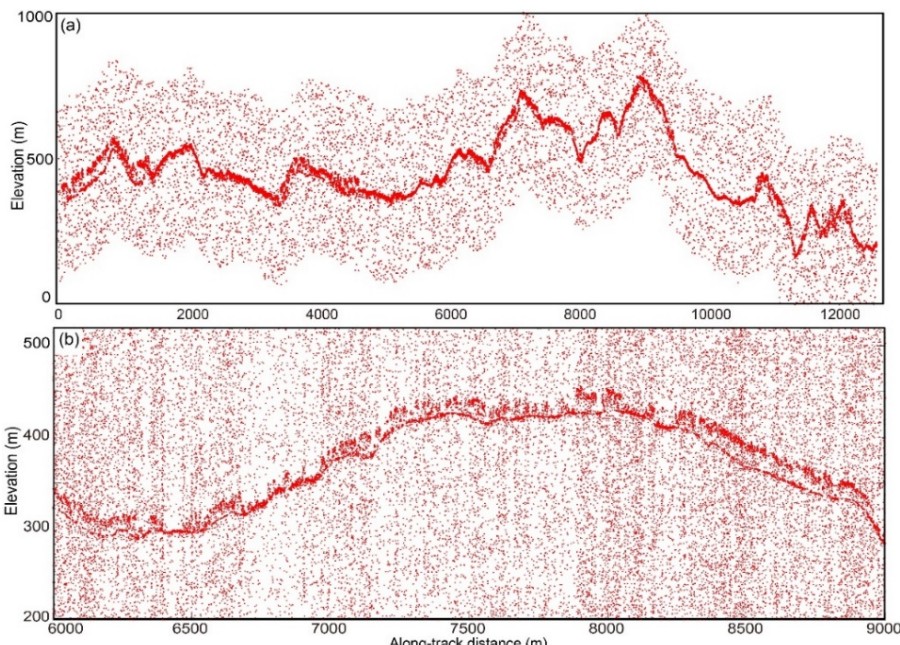

**Figure 1.** (**a**) Original photons of test site A. (**b**) Original photons of test site B.

### 2.1.2. ICESat-2 Dataset and Test Site

The proposed method was also tested using ATL03 photon cloud data acquired by the ICESat-2 system. The ATL03 data contain the height, latitude, longitude, and time [5]. Figure 2 shows the original photon maps of the spaceborne photon cloud dataset. Test site C is located in Africa, with a complex forest structure, which can test the effectiveness of the proposed method on ICESat-2 data well. The granule ID of these data is ATL03_20190414053207_02440301_001_01, and we obtained the ATL03 photon cloud data from the website (https://search.earthdata.nasa.gov/search, accessed on 15 October 2019).

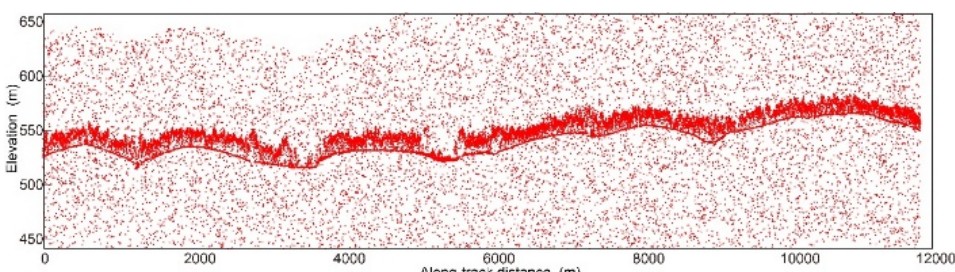

**Figure 2.** Original photons acquired by the ICESat-2 system.

### 2.1.3. Validation Data

With the auxiliary photon classification results provided by NASA, we carefully performed a visual inspection to add signal photons and remove noise photons [17]. Therefore, the obtained signal photons used to quantitatively evaluate the proposed method are classified.

## 2.2. Elliptical Distance-Based Filtering Parameters

It is a fact that the ellipse-based neighborhood search method is more suitable for finding the neighbors of a photon with similar distribution characteristics than the circle-based neighborhood search method [14]. In addition, to adapt to different terrain conditions, the adaptive ellipse-based neighborhood search method [21] was used to define the forward elliptical distance (*FED*) and backward elliptical distance (*BED*) [21]. Based on this, Yang et al. defined the backward local distance (*BLD*) to distinguish signal photons and noise photons adjacent to signal photons, thoroughly. However, photons are divided into three types, as shown in Figure 3, including signal photons, noise photons adjacent to signal photons, and noise photons far away from the signal photons; the single statistical parameter *BLD* cannot clearly distinguish three types of photon cloud. For example, although the parameter *BLD* shows better performance at distinguishing signal photons and noise photons adjacent to signal photons [21], it has a poor performance on retraining signal photons enough due to its sensitivity to noise photons. Therefore, the other two statistical parameters are defined to express the spatial density difference in photon clouds together with the *BLD*. The *FLD* is defined to better distinguish signal photons and noise photons far away from signal photons, and the neighboring forward local density difference (*NFLDD*) is defined to retain some signal photons that can be easily recognized as noise photons by the *BLD* parameter. Therefore, different types of photon clouds can be better expressed by three parameters rather than one parameter.

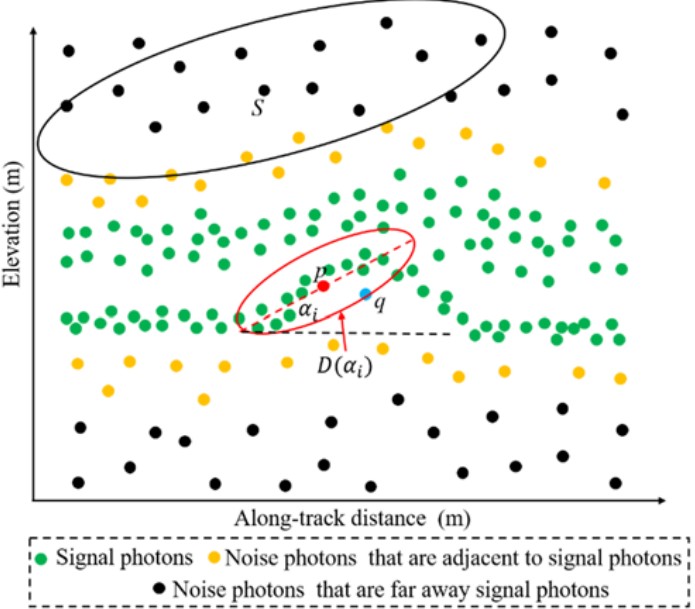

**Figure 3.** Schematic diagrams of the photon distribution and the forward elliptical distance.

### 2.2.1. Forward Elliptical Distance and Forward Local Density

The forward elliptical distance (*FED*) $D_{pq}(\alpha_i)$ from photon $p$ to its neighboring photon $q$ is defined as Equation (1).

$$\begin{cases} D_{pq}(\alpha_i) = \sqrt{\frac{\Delta x^2}{a^2} + \frac{\Delta z^2}{b^2}} \\ \Delta x = \cos \alpha_i \cdot (x_p - x_q) + \sin \alpha_i \cdot (z_p - z_q) \\ \Delta z = \cos \alpha_i \cdot (z_p - z_q) - \sin \alpha_i \cdot (x_p - x_q) \\ \alpha_i = \frac{i}{60} \cdot \pi, i = 0, 1, 2, ..., 60 \end{cases} \tag{1}$$

where $a$ and $b$ are the long and short semi-axes of the ellipse, respectively, which can be empirically set according to the surface cover, e.g., $\alpha_i$ is the clockwise angle between the

long semi-axis and the horizontal direction, which is used to define the orientation of the ellipse. The *FED* varies with the angle $\alpha_i$.

Based on the *FED*, the *FLD* of photon point $p$ is defined as in every orientation $\alpha_i$, the sum of the *FED*s is calculated, and the minimum sum value is regarded as the *FLD*:

$$FLD_p = min\left(\sum_{q=1}^{K_1} D_{pq}\left(\alpha_i^F\right)\right) \tag{2}$$

where $K_1$ is the number of nearest neighbors selected by the ellipse search window. The corresponding orientation $\alpha_i$ is called the forward local density orientation (*FLDO*) and is expressed as $\alpha_{min}$, which is the optimal search direction.

For example, for the two photons $p$ and $s$ in Figure 3, the *FLD* of signal photon $p$ is significantly different from that of noise photon $s$. However, for the noise photons adjacent to signal photons, since they distribute in a similar way to the signal photons, it is difficult to determine an appropriate threshold for *FLD*s to identify signal photons. To solve this problem, the backward elliptical distance (*BED*) was then defined [21].

### 2.2.2. Backward Elliptical Distance and Backward Local Density

The *BED* from photon $p$ to photon $q$ is defined as shown in Figure 4. In the *FLDO* $\alpha_{min}^q$ of photon $q$, the *FED* from $q$ to $p$ is called the *BED* from $p$ to $q$, as shown by $qp'$ [21]. The difference between the *FED* and the *BED* is the use of different *FLDO*s to calculate their elliptical distance values. The *BLD* of $p$ can then be calculated by the sum of the *BED* values corresponding to different orientations $\alpha_{min}^q$ [21]. It can be known from its definition that, for one photon, its neighboring photons with similar *FLDO* values are preferred for calculating the backward local density (*BLD*). Thus, for a noise photon that is adjacent to signal photons, the *BLD* does not tend to choose the signal photons to calculate its *BLD* [21]. Therefore, the *BLD* has the ability to distinguish signal photons and noise photons adjacent to signal photons.

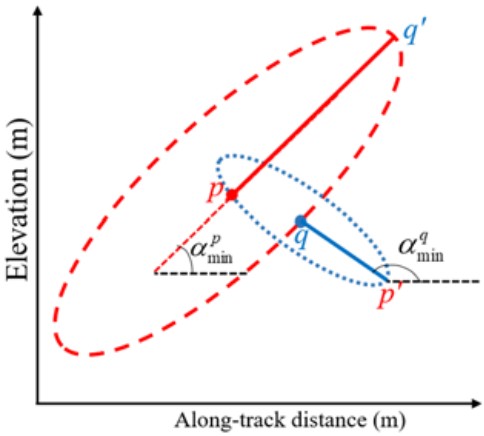

**Figure 4.** Schematic diagrams of backward elliptical distance [21].

### 2.2.3. The Neighboring forward Local Density Difference

For a forested area, the signal photons interacting with the dense canopy and ground can be easily recognized by the *FLD* and *BLD*. However, gaps occur between the canopy and ground surface and signal photons are sparsely distributed. As a result, these signal photons are easily misclassified into noise photons since their distribution characteristics are similar to those of the noise photons adjacent to signal photons. To overcome this problem based on the *BLD*, the neighboring forward local density (*NFLD*) is defined as follows:

$$NBLD_p = min\left(\sum_{q=1}^{K_1} D_{qp}\left(\alpha_{min}^q\right)\right) \tag{3}$$

where $K_1$ is the number of nearest neighbors selected by the ellipse search window. Before giving the definition of *NFLD* difference (*NFLDD*), we first divide the ellipse search window into two parts by a line *L*, as shown in Figure 5c, where line *L* is the perpendicular line of the long axis of the ellipse:

$$
\begin{cases}
y = x_p & \alpha^p_{min} = 0 \ or \ 180^\circ \\
y = 0 & \alpha^p_{min} = 90^\circ \\
y = -\dfrac{1}{\tan \alpha^p_{min}} \cdot (x - x_p) + y_p & 0 < \alpha^p_{min} < 90^\circ \ \cup 90^\circ < \alpha^p_{min} < 180^\circ
\end{cases}
\tag{4}
$$

where $x_p$ and $y_p$ are the coordinates of photon $p$ along the horizontal and vertical directions, respectively. The *NFLDD* is then expressed as follows:

$$
NBLDD_p =
\begin{cases}
\left| min(NBLD_{part1}) - min(NBLD_{part2}) \right| & case1 \\
\left| min(NBLD_{part1 \ or \ part2}) - min(NBLD_p) \right| & case2
\end{cases}
\tag{5}
$$

where the geometrical explanations of cases 1 and 2 are shown in Figure 5a,b. For signal photon $p$ between the canopy photons and the ground photons, since its neighboring photons selected by the ellipse search window are dominated by signal photons with similar distribution characteristics, it has a relatively small *NFLDD* value. However, for noise photons adjacent to signal photons, the two groups of neighboring photons are dominated by the signal and noise photons. Consequently, it has a relatively large *NFLDD* value. In such a case, the *NFLDD* can retain the signal photons and further enhance the ability to identify the noise photons adjacent to signal photons and the sparse signal photons.

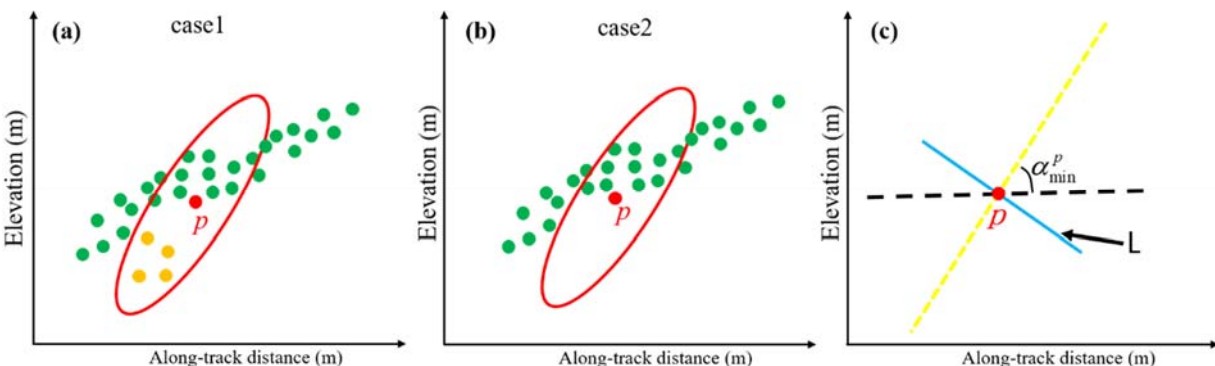

**Figure 5.** (**a**,**b**) geometrical explanations of (5). Schematic diagrams of (**c**) the ellipse divided into two parts by line *L*. Green dots represent signal photons; Yellow dots represent noise photons that are adjacent to signal photons.

### 2.3. Filtering Method Based on Machine Learning Approach

#### 2.3.1. Machine Learning Approach

The *FLD*, *BLD*, and *NFLDD* are used for describing the density distribution of photons in space. However, the problem that should be considered is how to use the three attribute parameters to remove the noise photons and retain as many signal photons as possible. A simple strategy adopted by the existing methods is to calculate the statistics of one attribute parameter and then fit the statistics by a specific probability density function, which is then utilized to remove the noise photons by setting a threshold. However, in this study, this process could not be easily carried out because three attribute parameters are used to describe the density distribution of the photons and predefining a reasonable probability density function and an optimal threshold for filtering the noise photons is difficult. Thus, based on the above three parameters, we use a machine learning method, namely, the

support vector machine (LiBSVM) method [25,26], to distinguish the signal photons from the considerable background noise photons.

### 2.3.2. Filtering Progress

Figure 6 shows the flow chart of the proposed method; there are three key procedures that should be carefully considered:

(1) Photon cloud coordinate system conversion: We convert the raw photon cloud data under the WGS-84 coordinate system to the Universal Transverse Mercator Grid System (UTM). The distance between all photon points and the starting photon is calculated along with the track distance (m); therefore, the elevation of photon cloud data is rearranged according to these distances.

(2) Coarse filtering: The coarse filtering procedure is used to remove the noise photons that are far away from the signal photons, which can improve the computational efficiency [9]. The main steps are as follows. First, the original photon cloud data are divided into multiple parts with a size of X meters along the flight direction and Z meters along the vertical direction. Subsequently, along the vertical direction, the number of photons in each grid cell is counted and the grid cell with the most photons is regarded as the grid cell containing the signal photons. To ensure that no signal photons are omitted, two or four neighboring grid cells of the selected grid cell along the vertical direction are also selected.

(3) Training sample selection: Before performing the filtering process, the noise photon samples far away from signal photons, the signal photon samples, and the noise photon samples adjacent to signal photons should be provided to find the relationship between the photon types and the proposed elliptical parameters (*FLD*, *BLD*, and *NFLDD*) by LiBSVM. In this study, we build the filtering model and assess the performance of the filtering method by cross-validation. In detail, for the three experiments, only 3–6% of the total photons are randomly selected for training, and the rest 94–97% of the total photons are used for testing the proposed method. To achieve this goal, the three kinds of photons are randomly selected by artificial classification. Regarding the selection of training samples, we roughly divide the photons into three types according to the histogram statistics of the three defined parameters of the photon cloud. In addition, some training samples are then randomly selected. These preliminarily selected samples inevitably have errors; therefore, we remove all the obviously wrong sample points through visual interpretation. Through the above steps, the training samples can be selected semiautomatically.

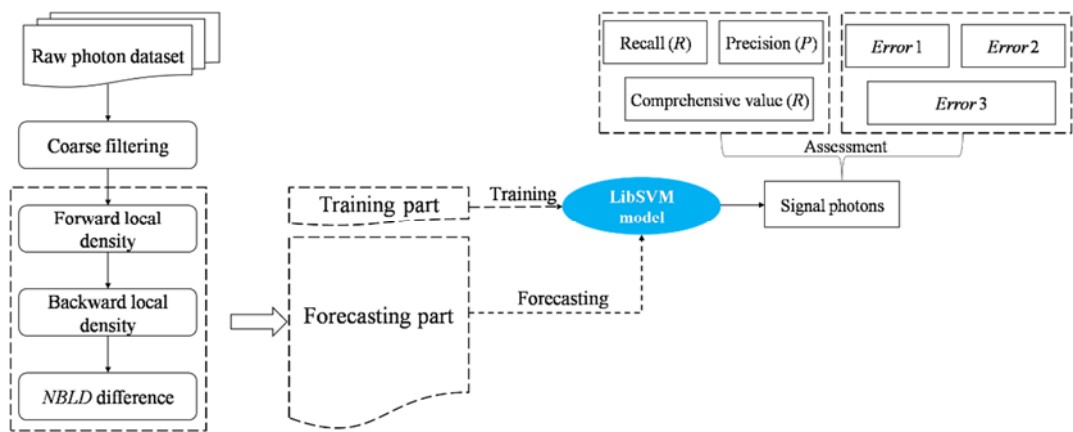

**Figure 6.** Flow chart of the proposed method.

### 2.3.3. Accuracy Indicators

To evaluate the performance of the proposed method, the recall *R*, the precision *P*, and the comprehensive evaluation value *F* are introduced [8,27], as is shown in Figure 4. *R*

indicates the ratio of the correctly identified signal photons to the total signal photons, $P$ is the ratio of the correctly identified signal photons to the total identified photons, and $F$ refers to the harmonic mean of the recall ratio $R$ and the precision $P$. In addition, another three error indices are also employed to evaluate the different methods [28]. The first error index $e_1$ denotes the ratio of the signal photons misclassified into noise photons to the total signal photons; the second error index $e_2$ is the ratio of the noise photons misclassified into signal photons to the total noise photons; and the third error index $e_3$ is used to express the ratio of all the misclassified photons to the total photons.

## 3. Results

### 3.1. Performance Assessment on Airborne Photon Cloud Data

The original photon cloud data in Figure 1 were firstly filtered, and the results are shown in Figures 7 and 8. Specifically, the long semiaxis and the short semiaxis of the ellipse in (1) were set as 15 m and 4 m, respectively. The number of neighbors (K) was set as 30. To better evaluate the performance of our filtering method, the LDS-based method [14], which has been widely employed to filter various photon cloud data [10], was also used to filter the photon cloud data. Due to the difference between the two datasets, the selected training samples accounted for 4.07% and 5.68% of the total photons in MATLAS and MABEL data. For both test sites, the proposed method can remove more noise photons than the LDS method. As a result, we can more easily identify the signal photons from the canopy layer and the signal photons from the ground surface, which helps us to extract the vegetation height and the sub-canopy topography.

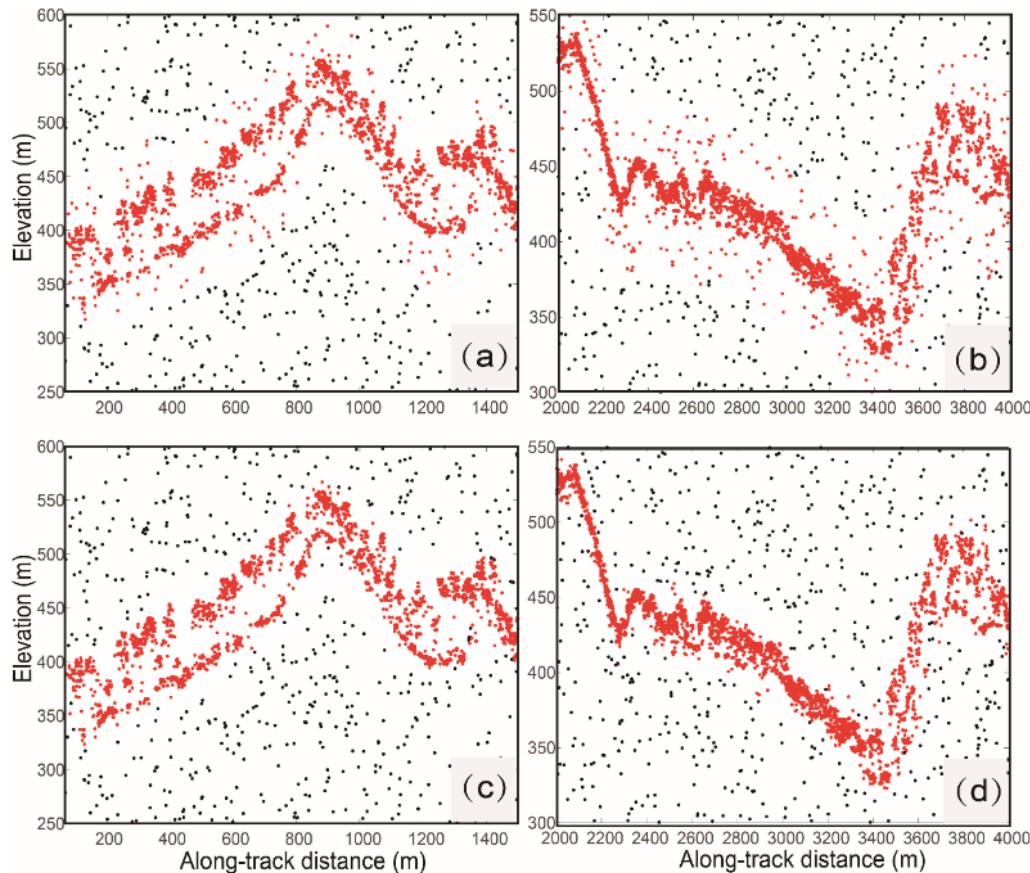

**Figure 7.** Partially enlarged views of test site A filtered by the LDS method (**a**,**b**) and the proposed method (**c**,**d**). Red dots represent signal photons; black dots represent noise photons.

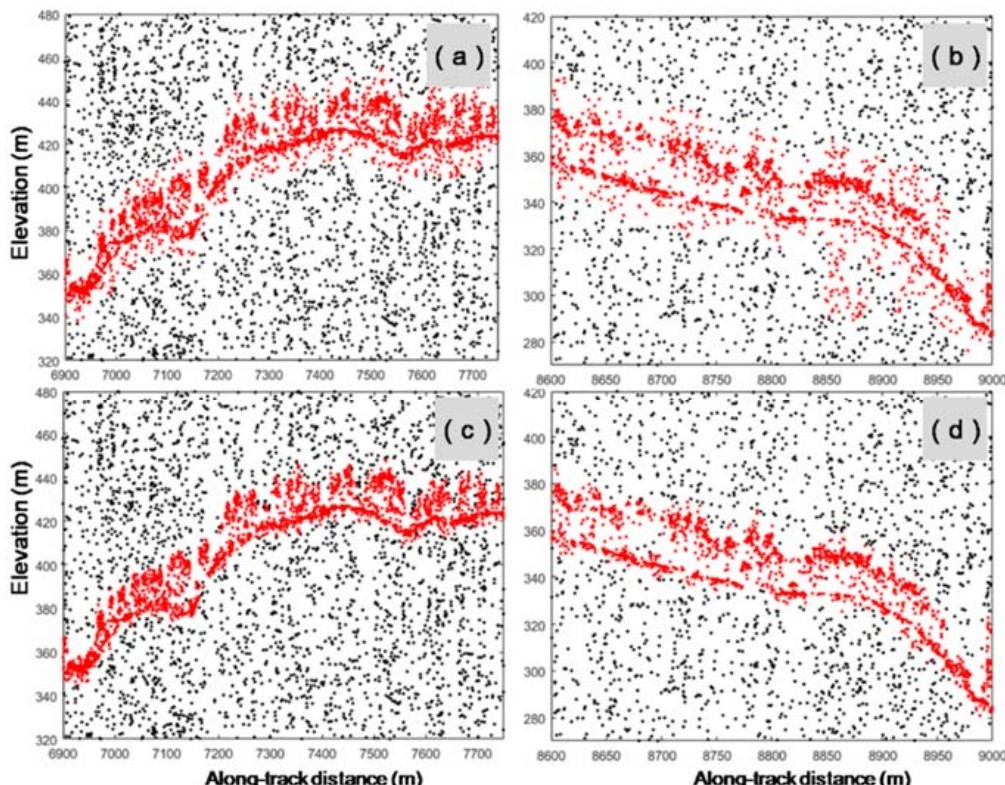

**Figure 8.** Partially enlarged views of test site B filtered by the LDS method (**a**,**b**) and the proposed method (**c**,**d**). Red dots represent signal photons; black dots represent noise photons.

To quantitively evaluate the performance of the above two methods in filtering photon data, three accuracy assessment parameters (*R* values, *P* values, and *F* values) and three error indices ($e_1$, $e_2$, and $e_3$) are utilized, as shown in Figure 9. For test site A, the accuracy assessment indicator values (*R*, *P*, and *F*) for the LDS method are 0.9995, 0.9560, and 0.9773, respectively, and the three error indices ($e_1$, $e_2$, and $e_3$) are 0.05%, 4.05%, and 2.18%, respectively. For the proposed method, the corresponding accuracy assessment indicator values are 0.9948, 0.9961, and 0.9954, respectively, as well as the error indices ($e_1$, $e_2$, and $e_3$) being 0.52%, 0.35%, and 0.43%, respectively. The recall *R* for the LDS method is slightly larger than that for the proposed method, and the error index $e_1$ for the LDS method is smaller than that for the proposed method, which means that more signal photons have been removed by the proposed method during the filtering process. However, this does not mean that the LDS method performs better in filtering noise photons than the proposed method, and although the LDS method can retain the signal photons, fewer noise photons are removed, as demonstrated by the precision *P* and the error index $e_2$. In particular, for the LDS method, 4.05% of the total noise photons are misclassified into signal photons, which is significantly more than for the proposed method. The comprehensive evaluation value *F* and the error index $e_3$ can provide a comprehensive evaluation of all the misclassified photons, and both indicate that the proposed method has the better advantage to balance any retaining signal photons and to remove as many noise photons as possible.

For test site B, although the photon cloud data are disturbed by the noise photons with a nonuniform distribution, a satisfactory result is still obtained by the proposed method, with the accuracy assessment indicators (*P* and *F*) being larger than 0.98. Figure 9 shows that the differences in recall *R* and error index $e_1$ associated with the LDS method and the proposed method are smaller than those in test site A, which means that for this test site, the two methods have a similar ability to retain signal photons. In addition, compared with the result for test site A, due to the impact of the dense noise photons with a nonuniform distribution at test site B, more signal photons are misidentified as noise photons.

Nevertheless, fewer signal photons are removed, which does not significantly distort the useful information in the signal photons. In terms of the other two accuracy assessment indicators and the two error indices, they all demonstrate that the new method shows a good performance in removing noise signals and retaining signal photons. Furthermore, we can conclude that the proposed method has a stronger tolerance to noise photons than the LDS method since the comprehensive evaluation value $F$ and error index $e_3$ do not decrease as much when the noise level is high.

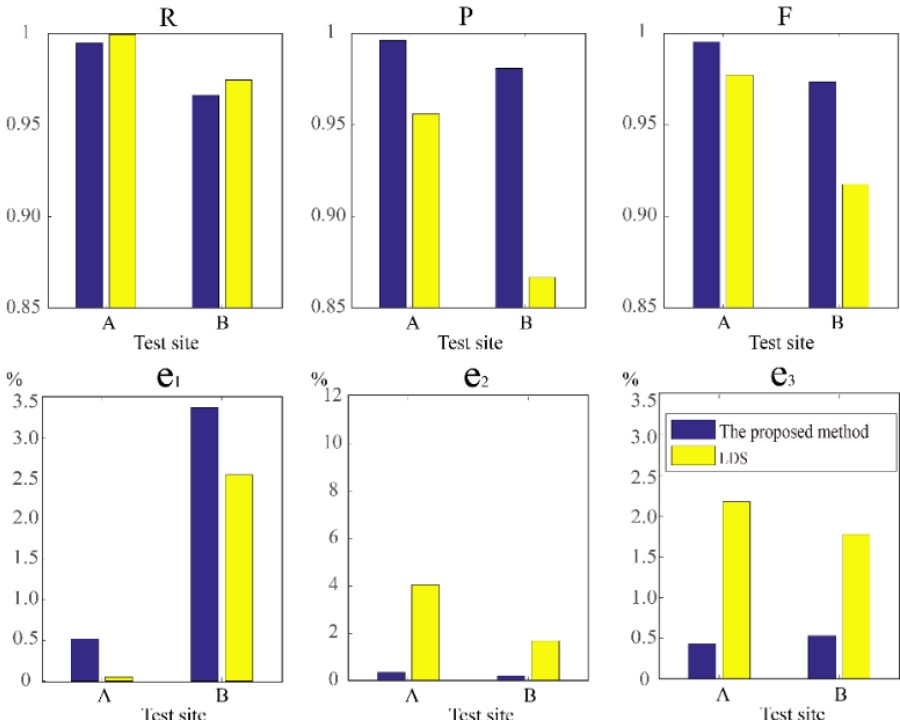

**Figure 9.** Accuracy assessment indicators and error indices corresponding to the results of the LDS method and the proposed method.

### 3.2. Performance Assessment on ICESat-2 Data

The original photon data are shown in Figure 2, which were also processed via the steps shown in Figure 6 and the LDS method. The selected training samples accounted for 3.62% of the total photons. To quantitatively evaluate the experimental results of the ICESat-2 data, we selected signal photons by visual interpretation. Table 1 shows that the accuracy assessment indicator values ($R$, $P$, and $F$) for the proposed method are 0.9977, 0.9886, and 0.9931, respectively, and the three error indices ($e_1$, $e_2$, and $e_3$) are 0.23%, 3.13%, and 1.01%, respectively, which are better than those of the classical filtering method. As is shown in Figure 10, the final results are presented, which show that the result of the LDS method has retained more noise photons than the proposed method. The reason is consistent with that for the airborne experiment, and we can conclude that the proposed method is also able to filter ICESat-2 data.

**Table 1.** Filter results of ICESat-2 data.

| Accuracy Indicators | The Proposed Method | LDS |
|:---:|:---:|:---:|
| $R$ | 0.9977 | 0.9963 |
| $P$ | 0.9886 | 0.9596 |
| $F$ | 0.9931 | 0.9776 |
| $e_1$ (%) | 0.23 | 0.37 |
| $e_2$ (%) | 3.13 | 11.29 |
| $e_3$ (%) | 1.01 | 3.33 |

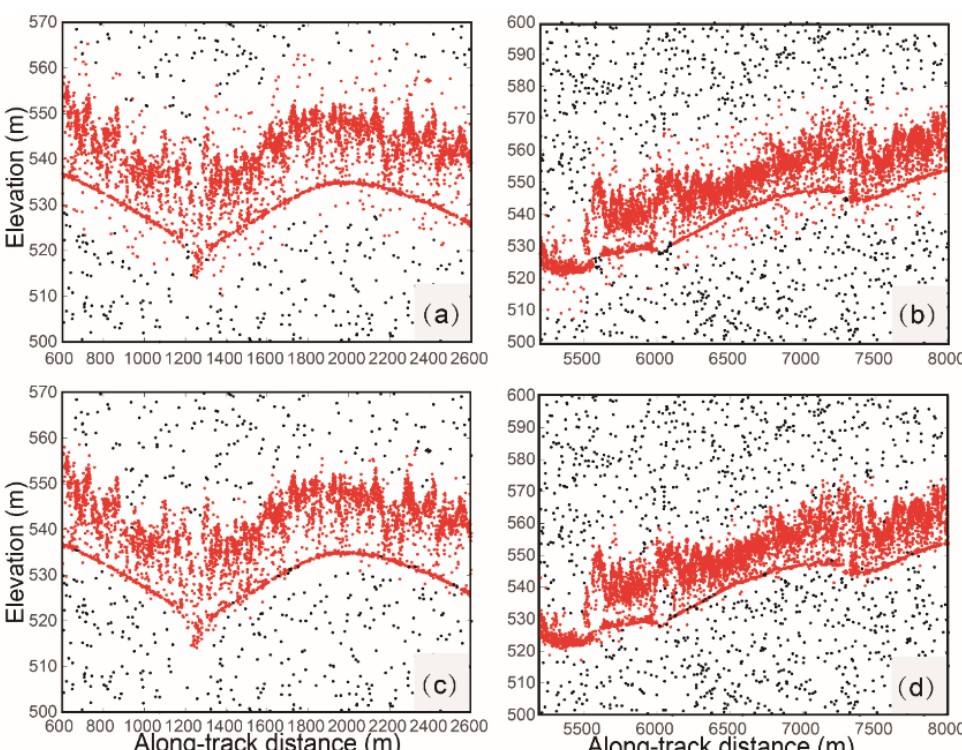

**Figure 10.** Partially enlarged views of the ICESat-2 data filtered by the LDS method (**a**,**b**) and the proposed method (**c**,**d**). Red dots represent signal photons; black dots represent noise photons.

## 4. Discussion

Although localized statistics-based filtering methods [13–22] have been widely used in forest areas, it is difficult to completely distinguish signal photons from noise photons. These localized statistics-based filtering methods [13–22] all use one statistical parameter to measure the density of the photon point cloud. Actually, the photon cloud can be divided into three categories: signal photons, noise photons far away from signal photons, and noise photons adjacent to signal photons. One statistical parameter can better distinguish two categories of photons at most. Moreover, it is difficult to find an optimal threshold for classifying the signal photons and noise photons adjacent to signal photons [19], because noise photons adjacent to signal photons present similar distribution trends as the signal photons. To better describe the distribution of photons, the *FED* and *BED* were used to model the spatial relationship between two photons. The *FLD*, *BLD*, and *NFLDD* attribute parameters were also defined to describe the density distribution of photon cloud data. A machine learning method, the support vector machine (LiBSVM) method [26], was used to combine the proposed statistical parameters.

The existing method only depends on one statistical parameter (such as LDS [14]) to describe the distribution of photons, which has a low sensitivity to the different distributions between signal photons and noise photons. In order to explain the reason behind this phenomenon intuitively, some signal photons and noise photons adjacent to signal photons were selected for analysis. The local distance in the existing method [14] and the three elliptical parameters in the proposed method were calculated for every selected photon. The corresponding statistical results for the above four parameters are represented by the box plots shown in Figure 11. The difference in the LDS between the noise photons and the signal photons is small (see Figure 11a), which explains the difficulty in distinguishing the signal photons and noise photons. Similarly, the *FLD* also shows a poor ability to separate noise photons from signal photons (see Figure 11b). However, the box plots (see Figure 11c,d) show that the *BLD* and *NFLDD* values corresponding to the noise and signal photons are significantly different. These attribute parameters both have a better ability to distinguish noise and signal photons. Since the photons can be classified into noise photons

far away from signal photons, noise photons adjacent to signal photons, and signal photons, their distribution is difficult to describe by only one parameter. The proposed *FLD*, *BLD*, and *NFLDD* attribute parameters provide us with the possibility to better understand the distribution of the photons.

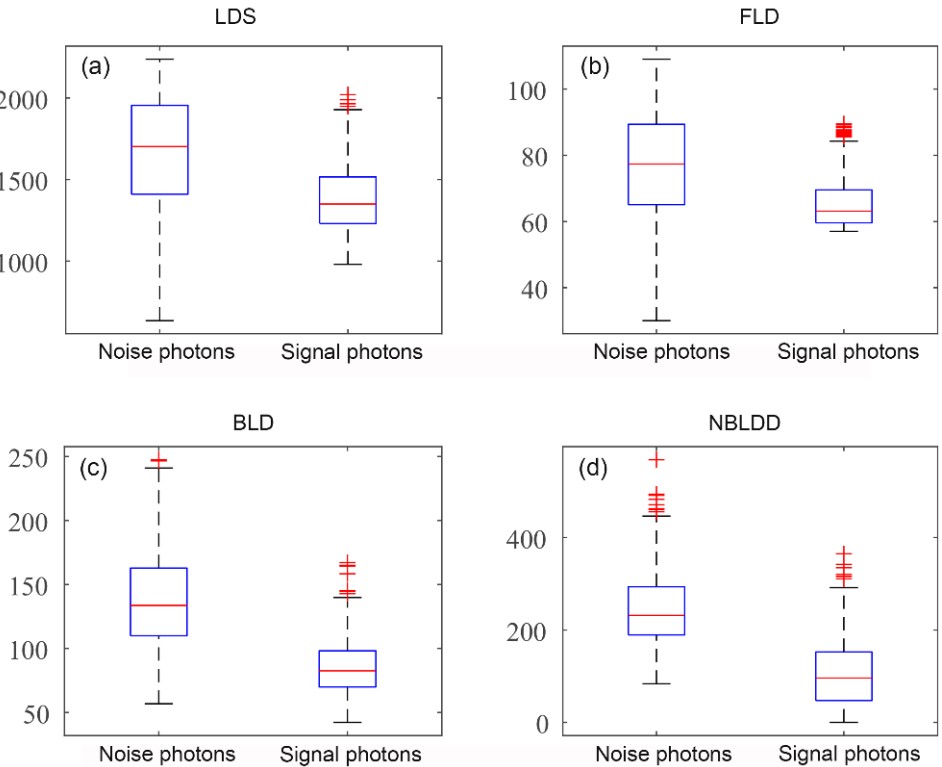

**Figure 11.** Local distance statistics results for (**a**) LDS, (**b**) *FLD*, (**c**) *BLD*, and (**d**) *NFLDD*.

## 5. Conclusions

Forest vertical structure parameters play an important role in carbon cycle investigations. The new generation of the spaceborne lidar (ICESat-2) was equipped with the ATLAS system, and has been demonstrated to be a potential tool for obtaining forest vertical structure parameters on a large scale. However, there are a lot of background noise photons in the photon cloud. Therefore, photon cloud filtering is a crucial step for obtaining forest vertical structure parameters.

In this paper, different from previous studies, the three statistical parameters are used to jointly measure the spatial density of the three types of photon cloud by a machine learning approach. The relevant assessment revealed that the three statistical parameters combined with the machine learning approach can accurately filter noise photons and retain signal photons accurately; most noise photons adjacent to signal photons were distinguished correctly. The performance of the proposed method showed that the three statistical parameters are easier to reflect the spatial density differences between the three types of photon cloud than the single statistical parameter. In addition, by combining the *FLD*, *BLD*, and *NFLDD* attribute parameters with the machine learning tool, it is possible to distinguish the noise and signal photons without depending on any statistical model or threshold. These findings demonstrated that the proposed filtering method would be useful for forest vertical structure parameter inversion of ICESat-2 vegetation and surface research. However, the performance of the machine learning-based filtering process depends on the number of training samples and their spatial distribution. In our future work, we will pay more attention to the effect of the training samples on the filtering performance and focus on the smallest possible number of training samples and their optimal distribution. In addition, the proposed method will also be tested using a photon cloud dataset acquired over other surfaces.

**Author Contributions:** Conceptualization, Y.L., J.Z. and H.F.; methodology, Y.L., J.Z. and H.F.; validation, Y.L., J.Z. and H.F.; formal analysis, Y.L., H.F. and S.G.; resources, Y.L., H.F., J.Z. and S.G.; data curation, Y.L., J.Z., H.F. and S.G.; writing—original draft preparation, Y.L.; writing—review and editing, Y.L., H.F. and C.W.; funding acquisition, H.F. All authors have read and agreed to the published version of the manuscript.

**Funding:** The work was partly supported by the National Natural Science Foundation of China (nos. 41820104005 and 41904004).

**Institutional Review Board Statement:** Not applicable.

**Informed Consent Statement:** Not applicable.

**Data Availability Statement:** The MABEL and MATLAS data are available from http://icesat. gsfc.nasa.gov/icesat2 (accessed on 15 October 2019). The ICESat-2 data are available from https: //search.earthdata.nasa.gov/search (accessed on 15 October 2019).

**Acknowledgments:** Special thanks to NASA for providing the datasets for free.

**Conflicts of Interest:** The authors declare no conflict of interest.

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
