# Peer review of "Filtering Photon Cloud Data in Forested Areas Based on Elliptical Distance Parameters and Machine Learning Approach"

_forests, doi:10.3390/f13050663_

Round 1

Reviewer 1 Report

Overall this paper is on a very interesting and important subject area. This paper studies the filter for the noise photons. I have a few major points for this study. In the literature review, the authors should provide a more comprehensive review of the existing studies related to noise photons.

The other comments are as follows

Comments

  • The needs of the Machine Learning Approach was not describe in the introduction .
  • Please elaborate on the findings and add the cross-validation statement to support your study.
  • Please add more references to support your statement. Add more latest references (Year 2018-2022)
  • Conclusions are too abundant and the main ideas are not fully extracted.
  • The conclusion should comprehensive with the findings from the study, the significant results and impact.
  • What is the novelty of this study? The author should highlight it in perfect academic writing style and thorough explanatory processing.

Reviewer 2 Report

In Introduction: "The proposed method has two main advantages."-compared to what method (in references or.....)?

In 2.3.3. what was the achieved accuracy?

The discussion should be performed also in comparison with previous studies. They should review the recent most important studies using similar spectral indices, and discuss how their findings support/vary from those previous findings.
